# Surface EMG in Subacute and Chronic Care after Traumatic Spinal Cord Injuries

## Gustavo Balbinot

KITE Research Institute, University Health Network, Toronto, ON M5G 2A2, Canada; gustavo.balbinot@uhn.ca

**Abstract: Background:** Traumatic spinal cord injury (SCI) is a devastating condition commonly originating from motor vehicle accidents or falls. Trauma care after SCI is challenging; after decompression surgery and spine stabilization, the first step is to assess the location and severity of the traumatic lesion. For this, clinical outcome measures are used to quantify the residual sensation and volitional control of muscles below the level of injury. These clinical assessments are important for decision-making, including the prediction of the recovery potential of individuals after the SCI. In clinical care, this quantification is usually performed using sensation and motor scores, a semi-quantitative measurement, alongside the binary classification of the sacral sparing (yes/no). **Objective:** In this perspective article, I review the use of surface EMG (sEMG) as a quantitative outcome measurement in subacute and chronic trauma care after SCI. **Methods:** Here, I revisit the main findings of two comprehensive scoping reviews recently published by our team on this topic. I offer a perspective on the combined findings of these scoping reviews, which integrate the changes in sEMG with SCI and the use of sEMG in neurorehabilitation after SCI. **Results:** sEMG provides a complimentary assessment to quantify the residual control of muscles with great sensitivity and detail compared to the traditional clinical assessments. Our scoping reviews unveiled the ability of the sEMG assessment to detect discomplete lesions (muscles with absent motor scores but present sEMG). Moreover, sEMG is able to measure the spontaneous activity of motor units at rest, and during passive maneuvers, the evoked responses with sensory or motor stimulation, and the integrity of the spinal cord and descending tracts with motor evoked potentials. This greatly complements the diagnostics of the SCI in the subacute phase of trauma care and deepens our understanding of neurorehabilitation strategies during the chronic phase of the traumatic injury. **Conclusions:** sEMG offers important insights into the neurophysiological factors underlying sensorimotor impairment and recovery after SCIs. Although several qualitative or semi-quantitative outcome measures determine the level of injury and the natural recovery after SCIs, using quantitative measures such as sEMG is promising. Nonetheless, there are still several barriers limiting the use of sEMG in the clinical environment and a need to advance high-density sEMG technology.

**Keywords:** spinal cord injury; electrophysiology; surface electromyography; trauma care

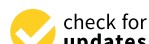



## 1. Introduction

Traumatic spinal cord injury (SCI) may affect the spinal cord's white and grey matters, leading to neuromuscular changes—which are reflected in the surface electromyography (sEMG) [1]. Trauma care after SCI relies on the understanding of the residual sensory and motor functions, which are used to classify the severity of the lesion using the impairment scale calculations of the American Spinal Injury Association (ASIA) via the International Standards for Neurological Classification of Spinal Cord Injury (ISNCSCI)—hereafter referred as ASIA impairment scale (AIS) [2–4]. In subacute trauma care settings, the quantification of the severity of the lesion is important for understanding the recovery prognostics and, respectively, prescribing the most appropriate rehabilitation therapy to the patient's needs [5,6]. The rehabilitation of individuals living with SCI can be divided into three phases: acute, subacute, and chronic. Although not consistently demarcated

in the literature, the acute and subacute phases combined generally correspond with the natural recovery period (12–18 months post-SCI), and the chronic phase is when the neurorecovery has plateaued [5,6]. Here, I will focus on how sEMG can be used as an assessment and prognostic tool in the subacute and chronic phases of SCI. For example, it is known that individuals classified as AIS A/B follow a worst recovery prognostic compared to individuals classified as AIS C/D early after the injury [7]. Nonetheless, a more detailed neurophysiological examination of the recovery potential of individual muscles may enhance the effectiveness of novel and promising therapies, which can be delivered to muscles with a potential for recovery [8].

sEMG is the most common electrophysiological assessment used in clinical trials after spinal injuries [9] and may complement the clinical testing by detecting the residual motor function in detail—including muscles with seemingly absent motor activities in the ISNC-SCI assessment [1]. While the ISNCSCI quantifies the residual motor command by using manual muscle testing (MMT) of relevant myotomes (scored using a semi-quantitative 6-point Likert scale), sEMG is a quantitative measurement not susceptible to ceiling or floor effects—which shows a good correlation with the MMT [10]. Thereby, sEMG is a promising tool for detecting the lesion severity and recovery prognostic early after the lesion, provided its great sensitivity. Additionally, sEMG may be used for understanding the effects of neurorehabilitation strategies at the chronic stages after the SCI, which many times are very subtle and demand more sensitive measurements.

Despite the quantitative and sensitive outcome measurements obtained from sEMG assessments, sEMG is still not broadly used and accepted in clinical practice. Indeed, provided the above-mentioned properties of the sEMG measurement, careful protocols for acquisition and analysis are needed. This demands qualified professionals such as kinesiologists and biomechanics experts, resources that are often unavailable in clinical settings [11]. This perspective article discusses the findings of two recent scoping reviews on the use of sEMG in SCI [10,12]. I revisit the combined findings of these scoping reviews, supporting the use of sEMG in trauma care after SCI. Finally, I discuss the opportunities for the use of sEMG in subacute and chronic trauma care after SCI and offer perspectives on how to advance this field.

## 2. Subacute Care: Monitoring and Classifying the Lesion Severity and Tracking of the Natural Recovery Process

In the acute phase of SCI, which usually refers to the first few days in intensive care after an injury, surgical decompression (within the initial 24 h) is associated with improved sensorimotor recovery; with the first 24–36 h post-SCI representing a crucial time window for obtaining optimal neurological recovery with surgery [13]. In the acute phase, several neurochemical biomarkers from the serum and cerebrospinal fluid are related to injury severity. These biomarkers show promise for stratifying injury severity and potentially predicting outcome [14–16]. Nonetheless, during the acute phase, sEMG is less useful for classifying injury severity and predicting long-term outcomes because of the period of spinal shock, which I will describe later in this article.

In the subacute phase of SCI, a pressing question early after the SCI is: "Am I going to fully recover from this injury?". Scientists and clinicians have been studying what subacute factors can be determinants of the recovery prognostic. Early after the traumatic injury, commonly, there is a period of spinal shock—in which many sensorimotor functions are lost. Soon after, some of these functions will re-emerge, and the amount of residual sensory and motor function is a long-known determinant of the recovery prognostic [17,18]. As previously mentioned, this is commonly quantified by the MMT and sensory tests used in the ISNCSCI in association with the AIS. With this information in their hands, the clinicians are able to answer the above-mentioned question with great confidence. For these professionals, this understanding is also very important because it may help in determining the best rehabilitation/treatment strategy for the patient. Nonetheless, more and more evidence point to the limitations of this classification [19], for example, very

different recovery profiles were evident in a group of individuals classified as sensorimotor complete SCI (AIS A) [20]. In this study [20], 22.3% of individuals classified with the most severe lesion using the AIS classification followed a good recovery trajectory (of the total motor score), whereas 54.2% had a moderate recovery trajectory and 24.5% had a marginal recovery trajectory [20]. This highlights the weaknesses of the AIS classification when predicting sensorimotor recovery after SCI. This may affect the patients' expectations about the recovery from the injury, with important psychological consequences, as well as the planning of the therapy by clinicians in subacute trauma care.

This is indicative that some nuances about the lesion severity and potential for recovery are not captured by the traditional clinical assessments. In 1992, Arthur Sherwood coined the term motor discomplete lesions when conducting sEMG assessments in a group of 88 clinically complete lesions, of which 74 (84%) were discomplete as defined by responses to the sEMG assessment [21]. This may indicate that the recovery prognostic may be better understood in terms of residual command to muscles if supplemented with information extracted from sEMG assessments. In our previous scoping review [1], we identified that 18/178 studies using sEMG assessments in SCI were conducted during the subacute phase of the lesion (<1-year post-injury). We identified that the sEMG assessment is capable of detecting the weakness of muscles [22,23] and the gradual recovery in neuromuscular activation [24]. Based on the reviewed literature, we were also able to identify gaps in the literature: although the solid relationship between the MMT (used in the ISNCSCI) and the sEMG assessment [10], the predictive value of the latter is still underexplored. Provided the great sensitivity of the sEMG assessment, it is reasonable to think that features obtained from the sEMG assessments early after the SCI would have a similar or enhanced predictive value compared to the clinical ISNCSCI assessment. This would contribute to a more assertive classification of the sensorimotor impairment after the injury, likely contributing to improvements in the classification of difficult cases [19] and sensorimotor complete SCI [20]. Indeed, our recent work indicated that neurophysiological biomarkers (i.e., the motor evoked potential) obtained in the subacute phase after SCI enabled enhanced prediction of muscle strength recovery [8]. This more assertive prediction of the recovery potential may allow the delivery of novel therapies, for example, anti-NOGO therapy [25] or paired associative stimulation [26], during the optimal time window for recovery after SCI.

After these acute and subacute phases, where the classification of the location and severity of the lesion is of importance, sEMG assessments can offer a more detailed description of the subsequent natural recovery. In Figure 1, I illustrate the quantification of the motor impairment and the natural recovery post-SCI using both the traditional clinical assessments of the AIS (e.g., motor score) and with the complement of sEMG assessments (alone or in combination with evoked potentials). In the period of spinal shock, strength, sensation, reflex activity, and sEMG amplitude may be absent for 14 weeks post-SCI; with the resolution of the spinal shock, tendon tap responses may reappear, and the sEMG amplitude may gradually increase (plateaus ≈ 22 weeks post-SCI) [27,28]. In these subacute phases, both the residual sEMG during volitional efforts or with brain or spinal cord stimulation are important factors for predicting the recovery potential of the individuals [8,29,30]. During the natural recovery process, the re-emergence of the reflex activity and signs of hyperreflexia and spasticity are also thought to be determinants of the recovery prognostic after SCI [29–31].

At the muscle level, muscles begin to atrophy within the first months after the SCI—evidenced by reductions in fiber diameter and cross-sectional area [32,33]. Evidence from animal models also indicates a shift in the proportion of muscle fiber types from fatigue-resistance (types I and IIa) to most-fatigable [types IIb and IIb(x)] (reviewed in [33]). This transition to a slower-to-faster fiber type begins after 4 to 7 months post-SCI and is continued chronically (reviewed in [31,33]). For example, in a cohort of individuals 10 months to 10 years post-SCI, ≈90% of the paralyzed muscles were dominated by type II muscle fibers [34]. Both the muscle atrophy and the increased fatigability may be detected

by sEMG assessments, with the use of sEMG amplitude and sEMG frequency (i.e., shifts in the median frequency), respectively.

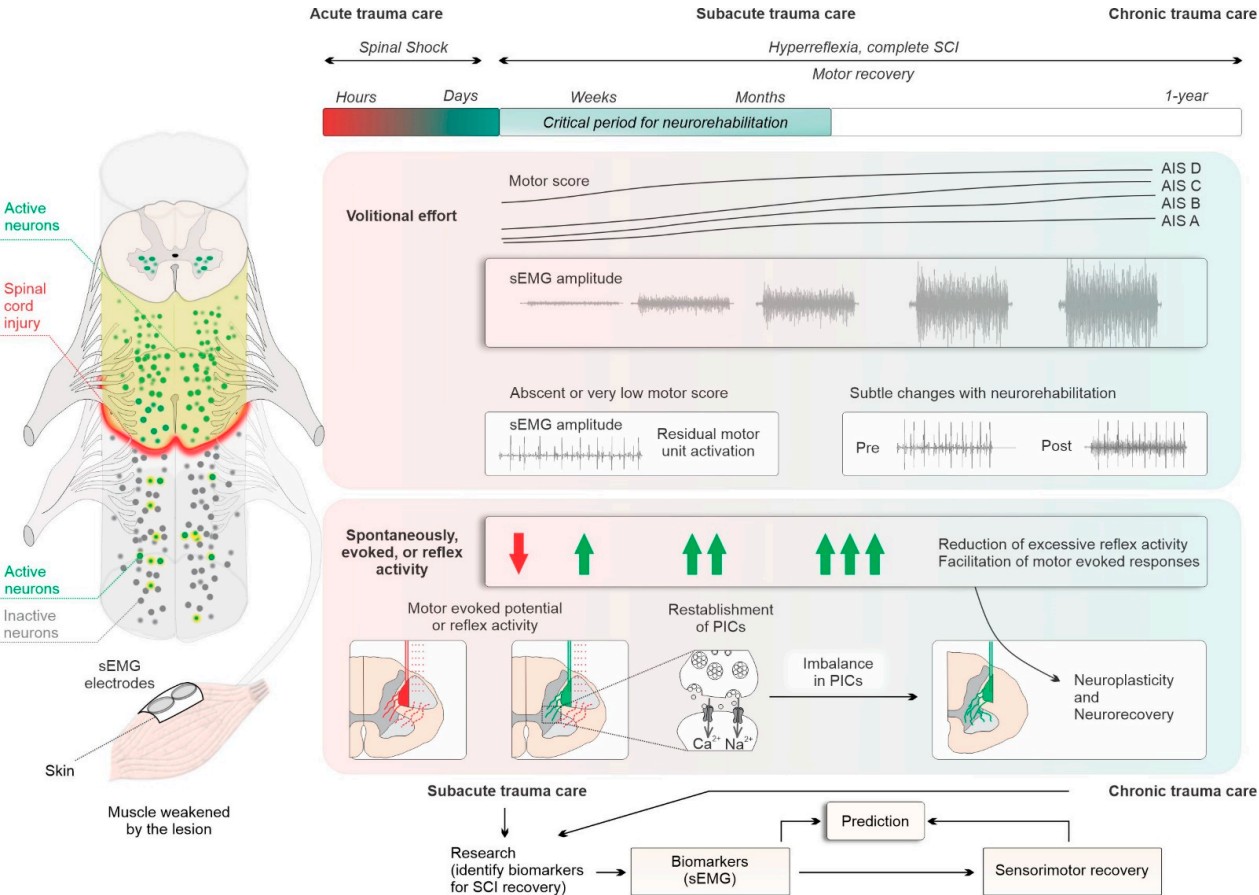

**Figure 1. The use of surface electromyography (sEMG) in subacute and chronic trauma care after spinal cord injuries.** (**Left panel**, from top to bottom) The spinal cord injury (red) may lead to damage to the grey and white matter of the cord, affecting the α-motoneurons (inactive neurons, grey). In order to produce a muscle contraction, residual neurons (active neurons below the level of injury, green) must be activated; nonetheless, the activation is often unable to produce a tetanic contraction and, respectively, produce force—in other words, a quantifiable motor score by the ISNCSCI. The use of electrodes on the skin overlying a muscle enables the detection of the residual activity of active motor neurons in detail (green neurons). (**Right panels**, from top to bottom) Acute, subacute, and chronic trauma care after spinal cord injuries under the perspective of the use of sEMG. In the acute phase, trauma management involves decompression surgery and stabilization of the injured segment. In the subacute period after SCI, after the period of spinal shock—where most of the sensorimotor and reflex activity is absent, there is a re-establishment of sensorimotor function and reflex activity. The more severe the lesion, the weaker the muscles. There is also the emergence of aberrant reflex activity (hyperreflexia). In the first months after the lesion, there is an upregulation of grown promoting factors, contributing to the critical period for neurorehabilitation. During this period, which usually encompasses the initial 6-months after the injury, most of the sensorimotor recovery occurs, evident in increases in motor scores and sEMG amplitude. sEMG assessments may add important information about the residual motor unit activation in muscles weakened by the lesion, especially in muscles with low motor scores (0–2). sEMG is also able to detect subtle changes with novel neurorehabilitation strategies. sEMG can quantify spontaneously, evoked, or reflex activity in muscles weakened by the lesion, contributing to a better understanding of the residual corticospinal projections and

the changes at the intrinsic spinal cord circuitry level; this includes the gradual increase in motor evoked potentials and the emergence of aberrant reflex activity, respectively—the latter thought to be mediated mostly by imbalances in persistent inward currents (PICs). During neurorehabilitation, sEMG may detect subtle changes such as the facilitation of motor evoked responses or the reduction of aberrant reflex activity (e.g., muscle spasms and spasticity). (**Bottom panel**) sEMG may be employed both in subacute and chronic trauma care to identify biomarkers for sensorimotor recovery of muscles—optimizing diagnostics and prognostics after the lesion, including the prediction of the efficacy of novel treatments. sEMG = Surface Electromyography; ISNCSCI = International Standards for Neurological Classification of Spinal Cord Injury; PIC = Persistent Inward Currents.

At the intersection between the muscle and the motoneuron lies the innervation zone of the muscles. Each motor unit consists of the $\alpha$-motoneuron and the muscle fibers innervated by this motoneuron. After an SCI, there is a redistribution of muscle innervation zones, which is possible to detect using sEMG assessments. For example, in individuals living with SCI, there is evidence of a wider range of innervation zones—if compared with a control non-disabled group [35]. These changes reflect the complex neuromuscular reorganization after the SCI, and sEMG may help in detecting these changes caused by the lesion and also the effects of novel treatments—which, for example, may act by promoting plasticity in the form of broadening of the reminiscent motor units' innervation zones.

At the motoneuron level, immediately after injury, the spinal cord enters a state of "spinal shock" characterized by severe muscle paralysis, flaccid muscle tone, and an initial loss of reflexes and sensation caudal to the lesion. In Figure 1, I also describe the underlying changes at the motoneuron level. The spinal shock period is initially characterized by the lack of volitional and reflex sEMG activity; with 1–3 days post-SCI, the H-reflexes begin to return, and later (4 days to 1 month), there is a gradual increase in all reflexes—which can be captured by sEMG (reviewed in [31]). One of the leading theories is that the main contributor to spinal shock at the motoneuron level is the disappearance of dendritic-voltage-activated sodium and calcium persistent inward currents (PICs) acutely after SCI. PICs amplify the synaptic response in motoneurons, leveraging the membrane potential to levels close to the threshold for firing; this allows fast and precise recruitment of motoneurons by volitional drive or sensory input. With the resolution of spinal shock, there is a recovery in the excitability of motoneurons, evidenced by increases in residual muscle strength (motor score) and sEMG amplitude, H-reflexes, F-wave persistence, and the return of reflex responses [27,28,31,36,37]. There is also a gradual recovery of the motor evoked potential [38] (this aggregated neurophysiological recovery is represented by green arrows in Figure 1). Motoneuron PICs re-emerge in the weeks after the SCI and contribute to motor recovery and also the development of spasticity and involuntary muscle and unit spasms [31]. The latter is a remarkable characteristic of the muscles weakened by the lesion: the spontaneous motor unit discharge discussed in our scoping review [1]. In brief, the deprivation of supraspinal efferences may lead to an imbalance of synaptic transmission in the intrinsic spinal cord circuits, leading to an increase in spontaneous lower motor neuron activity. For example, in SCI, passive knee movements can occasionally induce phasic sEMG activity and spasms [39], a phenomenon linked to the abnormal reflex responses [40–47] seen at the more chronic stages of the SCI—discussed in more detail in relation to PICs in the next Section 3. Chronic care: tracking persistent impairments and the effects of neurorehabilitation.

These responses may be detected using sEMG and serve as a biomarker, indicating the recovery prognostic of muscles weakened by the lesion. Nonetheless, more studies are necessary to understand the predictive value of these sEMG biomarkers for recovery after SCI. The above-mentioned electrophysiological biomarkers indicate the complex changes after the SCI, the subsequent plasticity in the intrinsic spinal cord circuitry, or at the innervation zones of muscles. Thereby, sEMG can add important information in subacute trauma care after SCI, with importance for diagnostics and prognostics of sensorimotor recovery.

Future studies should use sEMG to better understand the nuances of the residual upper motor neuron projections, the respective imbalance between motor efferents (absent or diminished) and sensory-afferents (which are fully preserved), the intrinsic changes in spinal cord excitability, and the changes at the innervation zones. In the next section, I will discuss the use of sEMG as a prognostic and assessment tool in chronic trauma care, focusing on the ability of sEMG to predict the response and detect the effects of novel and promising neurorehabilitation strategies in SCI.

## 3. Chronic Care: Tracking Persistent Impairments and the Effects of Neurorehabilitation

Persistent impairments are common after SCI, and there is a continuous effort from clinicians and scientists to improve chronic care for individuals living with SCI. Currently, there are no effective therapies available for the treatment of SCI. As reviewed in the last section, the sensorimotor recovery process depends on how serious the injury was and on the amount of cell loss, which are critical for sending signals from the brain to the rest of the body. In the more chronic stages of trauma care after SCI, rehabilitation therapy is used to increase muscle strength and bodily functions, and there is a constant effort to discover novel ways to maximize its effects by dosing or combining novel treatments. In this regard, sEMG can be used to quantify the effects of these novel neurorehabilitation strategies in detail—bridging the neuromuscular effects of the treatment to the possible functional gains with the therapy. Because there is no cure for SCI, many treatments may only induce subtle changes in sensorimotor control, which often may not lead to a significant functional improvement. Nonetheless, the understanding of the efficacy of novel treatments, no matter how small the change, may help to decide what treatments will be further explored. In this line of thought, sEMG may help to expand the knowledge and optimize the delivery of novel treatments to individuals living with an SCI. The need for more rehabilitation time to improve functional outcomes is a consensus between physiotherapists. In subacute trauma care, inpatient rehabilitation is the most important driver of direct health care costs after neurological injuries, and the resulting pressures on the healthcare system are leading to progressively shorter hospital stays. In chronic trauma care, rehabilitation programs need to be optimized and tailored to each individual in an evidence-based manner to ensure that the best possible use is made of the limited therapy time. Thereby, in both subacute and chronic scenarios, sEMG assessments may be used to enhance the prediction of treatment efficacy and quantify subtle neuromuscular changes with therapy.

We have identified that the field of neurorehabilitation after SCI would benefit from the addition of sEMG to the routine clinical assessments of impairment and recovery [12]—which, as previously mentioned, are usually conducted using the AIS. This stems from the fact that the MMT assessments used in the AIS classification lack the sensitivity to detect fine changes with novel neuromodulation treatments. Specifically, we identified that motor score changes from 0 (no strength) to 1 or 2 would benefit from the more detailed and quantitative sEMG assessment to detect sensorimotor recovery with novel treatments. Currently, changes in the scoring system are being proposed in the ISNCSCI and the Graded Redefined Assessment Of Strength, Sensibility, and Prehension (GRASSP) to account for these fine differences.

Thereby, the greater sensitivity of the sEMG assessment may be particularly important when there is an absence or a very weak volitional drive to muscles. This stems from the possibility of detecting sEMG activity in the absence of volitional drive (spontaneous activity), for example, in muscles with a motor score of 0, and detecting very fine changes in the motor unit recruitment in muscles with motor score change from 0 to 1 or 2. Such changes are often difficult to quantify by employing the MMT assessment. A recent study demonstrated how high-density sEMG could detect myoelectric activity and motor unit properties even in the absence of visible motion (motor score of 0) [48]. The study by Ting et al. (2021) [48] sets a benchmark for the use of high-density EMG to extract motor unit activity in SCI, which will allow clinicians to monitor neurorehabilitation over time and tailor them as necessary. In this study, the authors have demonstrated that measures

of motor unit function, such as discharge rate, interspike interval, and amplitude, can be obtained from high-density sEMG [48]. Importantly, motor unit activity can be extracted from muscles with low levels of sEMG-or even absent sEMG at volitional effort, which allows the understanding of motor recovery at the motor unit level (increase in active motor units) and motor unit dysfunction in pathologies such as spasticity [48]. Many studies have also taken advantage of sEMG assessments to understand the effects of neuromodulation or pharmacological treatments in reducing spontaneous motor unit activity (reviewed in [12]). These studies may employ the sEMG assessment of stretch reflexes during passive maneuvers—which are related to spasms and spasticity. Finally, sEMG has been used with careful protocols to detect the strengthening of corticospinal synaptic transmission and corticospinal-motor neuronal plasticity [49,50]. These examples illustrate the great sensitivity of the sEMG assessment in detecting the residual motor command and the valuable opportunity for testing the efficacy of novel treatments at the motor unit level.

At the chronic stages of SCI, sEMG can provide objective measurements of spontaneously occurring muscle activity, such as muscle spasms [51–54]. As mentioned earlier, motor units may show firing at rest or contraction-induced firing soon after volitional muscle activation—also called unit spasms [55,56]. These features are only captured by detailed neurophysiological assessments, highlighting their importance, especially in unresponsive muscles (absent or very weak contraction). The unit spasm is a subtle phenomenon not manifested in a visible or clinically measurable muscle spasm (Figure 2) [52,57]. Nonetheless, it is possible to detect the unit spasm by employing the sEMG assessment. This phenomenon is under investigation, and, as previously mentioned, the involvement of PICs is considered one of the leading theories in the field to explain its occurrence. The imbalance between cortical efferents (reduction of absence) and afferents arising from the periphery (maintenance or upregulation) caused by the SCI may lead to a long-term dysregulation of PICs—which may be detected by the sEMG assessments [31]. Recent evidence indicates that, potentially, non-pharmacological interventions (e.g., electrical stimulation) can attenuate unwanted PIC-induced muscle contractions [58]—which would explain the positive effects of electrical stimulation to enhance function [59,60] and reduce spasticity in SCI (reviewed in [12]). Thereby, with the advent of sEMG analysis, it is possible to identify nuances in the activity of the neuromuscular system, which can be later targeted by novel and promising treatments with the ultimate long-term goal of enhancing function in individuals living with SCI.

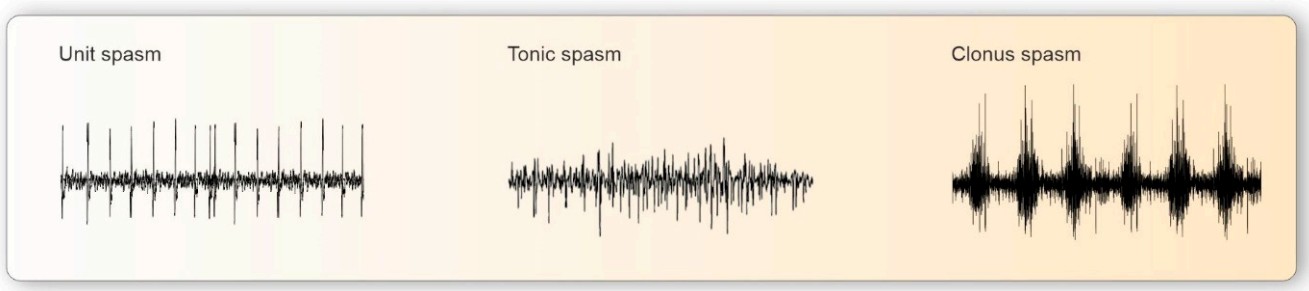

**Figure 2.** Spasm can be classified as unit, tonic, or clonus.

As previously mentioned, because there is no cure for paralysis, current treatments may result only in modest changes in neuromuscular function, which must be detected to provide initial evidence of the efficacy of the experimental treatment. This will allow further development of these novel treatments, with respective optimization or the combination with other treatment strategies—with the ultimate goal to improve function in individuals living with an SCI. We identified that studies on brain stimulation, locomotor training, spinal cord stimulation, and different strategies of pharmacotherapy have been employing sEMG to understand the changes in muscle activation during volitional efforts or the reduction of spontaneous muscle activity [12]. Additionally, the characterization of

volitional sEMG activity in motor complete SCI can be potentially used as a neuroprosthetic command source to compensate for full loss of movement in other muscles [48,61]. Finally, sEMG can also be used as a neurophysiological approach to nerve transfer to restore upper limb function in SCI [62].

## 4. Future Directions and Limitations

Further studies are needed to understand the predictive value of the different sEMG properties in prognosticating the recovery potential of individual muscles during the natural recovery process or with neurorehabilitation. For example, we have recently unveiled the importance of the motor evoked potential in predicting the strength recovery of hand muscles [8]; and we are currently investigating the ability of the sEMG assessment in predicting the responsiveness of muscles to functional electrical stimulation therapy. Importantly, sEMG is easier to use in the clinical environment, despite the barriers still limiting its use [11], in comparison to the transcranial magnetic stimulation assessments (used to obtain the motor evoked potential). Finally, more studies are necessary to understand the reliability of these neurophysiological measurements [63,64]; in the interim, rigorous protocols for acquisition and analysis alongside the use of control groups are highly recommended.

A limitation of this perspective article is the focus on sEMG studies conducted on individuals living with SCI. Although the field of sEMG is advancing in terms of technological enhancements—particularly on the use of high-density sEMG—most of the studies are still limited to non-disabled individuals. Such technologies will afford a better understanding of the changes in innervation zones and motor unit activity following the SCI, which is promising for prognosticating sensorimotor recovery and understanding the effects of novel neurorehabilitation strategies in the future.

## 5. Conclusions

In this perspective article, I discussed the use of surface electromyography in subacute and chronic trauma care after traumatic spinal injuries. Novel biomarkers obtained from this neurophysiological assessment will lead to a more assertive classification of the traumatic injury and of the potential for recovery of individual muscles with neurorehabilitation strategies.

**Funding:** This work was supported by the Wings for Life Spinal Cord Research Foundation (Project #210).

**Institutional Review Board Statement:** Not applicable.

**Informed Consent Statement:** Not applicable.

**Data Availability Statement:** Not applicable.

**Acknowledgments:** I would like to thank Jose Zariffa for important discussions contributing to the intellectual content presented in this article.

**Conflicts of Interest:** The author declares no conflict of interest.

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
