# Peer review of "Surface EMG in Subacute and Chronic Care after Traumatic Spinal Cord Injuries"

_traumacare, doi:10.3390/traumacare2020031_

Round 1

Reviewer 1 Report

Thank you for permitting me to review this manuscript 

On several area , We is used , since there is only one author this should be changed to I or rephrased otherwise 

A brief summary concerning , acute , subacute and chronic phase would be helpful for the reader to have a take home message.

Figure 1needs a better resolution 

normal EMG activity in healthy territories should be also availble for the average reader not specialized in EMG

Line 265-268 , please add some more details about this specific study 

 Line 282 please provide an EMG  representation of unit spasms

If possible please enumerate possible treatements which may partially affect SCI symptoms even very small partial effect 

Author Response

We would like to thank this reviewer for the comments that helped to improve the quality of the manuscript. Please note that all the changes in the manuscript are in track changes mode.

_Please note I have changed “we” to “I” in all instances referring to the present manuscript. Please note that in sentences referring to our previously published scoping reviews, I have kept the word “we”. I have also kept the word “we” on pp. 6, lines 270-275 to reflect the intellectual contribution of my peers to describe the ongoing changes in the manual muscle score quantification.

_I have added a description of the different phases of the SCI (acute, subacute and chronic) with the appropriate references. Please see these changes on pp. 2, lines 51-56.

_Figure 1 may display in lower resolution in the version for reviewers but was delivered in high quality (600 x 600 dpi). The published version should be in high resolution. I have improved the Figure to enhance visualization.

_I have added more information about the referenced study, please see this addition on pp. 7, lines 286-291.

_I have added Figure 2 to illustrate the unit spasms. I have also added references on this topic.

Winslow J, et al., IEEE J Biomed Health Inform. 2015.

Aguiar SA, et al., J Neurophysiol. 2018.

_I have described some of the treatments taking advantage of sEMG as an assessment tool. Please see this change on pp. 7, lines 325-328.

Reviewer 2 Report

The current manuscript aimed to discuss the available literature regarding the ability of surface EMG assessment in detecting the level of injury in SCI patients. In fact, different qualitative outcome measures such as ASIA to determine the level of injury, however, using quantitate measures such as EMG is really promising. I have some comments regarding this paper.

Abstract:

The study type is not clear, the author stated “in this perspective article” then “Our comprehensive scoping review”.  In addition, the abstract structure is not organized! it should include a brief introduction including the aim of the study, brief methods and results, and a clear conclusion. 

I suggest the author re-write the abstract again.

Introduction

Page 1, lines 30-39, Please cite the sentences appropriately.

The rest of the manuscript was written well, I suggest including some limitations. 

Author Response

We would like to thank this reviewer for the comments that helped to improve the quality of the manuscript. Please note that all the changes in the manuscript are in track changes mode.

_Abstract: I have re-organized the abstract in the following sections: background, objectives, methods, results, and conclusion. I have clarified in the abstract that this is a perspective article revisiting the main findings of two recent scoping reviews from our team. Please see these changes on pp. 1, lines 7-38.

_Introduction: I have cited the following references

Balbinot G, et al. J Neuroeng Rehabil. 2021. 

Committee A and IscIS. Spinal Cord. 2019. 

Furlan JC, et al. J Neurotrauma. 2008. 

Furlan JC, et al. J Neurotrauma. 2011. 

Kirshblum, et al. Archives of physical medicine and rehabilitation. 2007.

Hupp M, et al., J Neurotrauma. 2018. 

_I have included a limitations paragraph, please see this change on pp. 8, lines 346-352.